# Psychosocial Treatment of an Adolescent in a Hospital-Based Violence Intervention Program

**DOI:** 10.3390/children10061018

**Published:** 2023-06-05

**Authors:** Matthew Krock, Kimberly Burkhart, Edward Barksdale

**Affiliations:** 1Antifragility Initiative, Rainbow Babies and Children’s Hospital, 10524 Euclid Ave., Cleveland, OH 44106, USA; matthew.krock@uhhospitals.org; 2Department of Pediatrics, Rainbow Babies and Children’s Hospital/Case Western Reserve University School of Medicine, 10524 Euclid Ave., Cleveland, OH 44106, USA; 3Pediatric Surgery, Rainbow Babies and Children’s Hospital/Case Western Reserve University School of Medicine, 11119 Bellflower Rd., Cleveland, OH 44106, USA

**Keywords:** hospital-based violence intervention programs, psychosocial treatment, trauma

## Abstract

Interpersonal violence is the leading cause of death in adolescents. Hospital-based violence intervention programs (HVIPs) address the emotional and behavioral sequelae of assault and homicide. Few studies have been published on pediatric HVIPs, and no study offers a conceptualization model for treatment approaches. This paper demonstrates the use of empirically supported practices by outlining the services provided and subsequent treatment trajectory of an adolescent receiving care from an HVIP at a Level 1 Pediatric Trauma Center. This case study follows the Case Study Report (CARE) guidelines and is the first to demonstrate the use of intensive case management, Psychological First Aid (PFA), and Skills for Psychological Recovery (SPR) in the treatment of an adolescent presenting to the emergency department (ED) due to assault associated with bullying. Through the use of these treatment approaches to address symptoms of posttraumatic stress, the patient moved from physical recovery to posttraumatic growth. Assessment and trauma-informed treatment suggestions are provided to support evidence-based practices within HVIPs.

## 1. Introduction

In the United States, interpersonal violence is a major public health problem that has been declared a national endemic. It is the leading cause of death for individuals of color ages 15 to 34 [1]. Emergency departments (EDs) are often the first point of medical contact for youth after experiencing interpersonal violence. Violent injury accounts for approximately 600,000 adolescent emergency department visits each year [1]. Annually, these injuries result in more than 4 billion USD for acute medical care and 32 billion USD in lost wages [2]. Within the United States, 25% of high school age students report involvement in fighting in the past 12 months and 18% report carrying a weapon in the last month [3]. Hospital-based violence intervention programs (HVIPs) were created to address the emotional and behavioral sequelae of assault and homicide. HVIPs, which operate out of EDs and trauma centers, mitigate the risk of violent re-injury and perpetration by de-escalating potentially dangerous situations and supporting survivors of peer and community violence as they recover. Frontline workers build close relationships with survivors while providing a wide range of individually tailored services such as case management, advocacy, and trauma therapy [4].

Urban EDs in particular offer an opportunity to connect disenfranchised youth to behavioral health resources. Few studies investigating the impact of HVIPs on decreasing posttraumatic stress symptoms while fostering posttraumatic growth have been conducted. Even fewer studies have been conducted in pediatric HVIPs [5]. In one study conducted by Carter et al., a brief universal behavioral intervention was delivered at the bedside in the ED. The results indicated that this brief intervention, which combined motivational interviewing and cognitive skills training, was effective at reducing the frequency of violent aggression 2 months post-intervention. There were no changes, however, in rates of victimization [5]. No best practice model exists for treating victimized adolescents presenting to an ED. There is great need for a standardized conceptualization model and treatment approach that provides flexibility and can be applied in a variety of ED settings.

Key factors associated with greater community violence are increased exposure to adverse childhood experiences such as poverty, child abuse, structural racism, and parent incarceration and severe mental illness [6]. Community violence victimization is associated with posttraumatic stress symptoms, depression, delinquent behavior, increased use of substances, risky sexual behavior, lower school bonding, and involvement in the juvenile justice system. In a meta-analysis of 118 studies, self-regulation, family support, school support, and peer support were identified as having significant buffering effects against community violence [7]. This suggests that an intervention program delivered in response to youth victimization needs to include resources that address social determinants of health and culturally appropriate treatment intervention. The HVIP described in this paper is referred to as the Antifragility Initiative (AI). The foundation of AI is built on posttraumatic growth theory, which posits that there can be positive change after the experience of a traumatic event. Posttraumatic growth is the process of change in which individuals develop beyond previous levels of functioning. It closely aligns with a positive psychology approach in which the interplay between positive and negative life experiences helps us to flourish. Based on this theory, posttraumatic growth is measured by increased appreciation of life, relating to others, exploration of new possibilities, personal strengths, and spiritual change [8]. Therefore, the goal of the AI program is to provide services that promote posttraumatic growth, or in other words, positive transformation that can occur in the wake of trauma and adversity.

The AI program serves approximately 100 youth annually, with an average patient age of 15. The most common incident types treated in this program by order of frequency are assault, gunshot wound, and victim of homicide. These services include, but are not limited to, crisis intervention, intensive case management, trauma-informed therapeutic interventions, and the provision of essential goods and services. While HVIPs share a common mission, standardized practices are still in development. This paper fills an important gap in the literature by providing a conceptualization model for pediatric HVIP intervention and by demonstrating how empirically supported practices can be delivered within the context of the model. This case study is intended to serve as a guide for pediatric HVIPs by outlining the services provided and subsequent treatment trajectory of an adolescent receiving care from an HVIP operating out of a Level 1 Pediatric Trauma Center in the Midwest.

## 2. Procedure

Within the AI model, licensed independent social workers are present within the ED. Patients are identified by either the ED status board or directly approached by clinical staff. The AI social worker approaches the family and provides resources from the National Child Traumatic Stress Network (NCTSN) on types of victimization, signs and symptoms of trauma, and strategies to aid in recovery. Additionally, resources are provided to apply for aid through Victims of Violent Crimes (VOCA), as well as information on local resources to connect patients to social services. The social worker provides an overview of the AI program. Consent/assent to participate and treat is obtained. A psychosocial assessment is conducted over the course of 2 one-hour sessions. During the assessment, the intensity, frequency, and duration of posttraumatic symptoms is assessed, as are risk and protective factors. Sessions can occur in a hospital outpatient setting, the family’s home, or within the community (e.g., school; library). At the end of the second assessment session, a treatment plan is developed. One-hour sessions are provided weekly or bi-weekly. This is determined by both clinical judgment and the family’s availability. Services are provided until discharge criteria are met as defined by the patient’s treatment plan and based upon data obtained throughout the course of intervention or one-year post initiation of treatment. Should the patient require additional intervention after 1 year of services/intervention, the patient is referred to a partnering community mental health agency.

## 3. Case Presentation

### 3.1. Therapeutic Context

The patient is a 12-year-old multiracial female in the 8th grade who presented to a Level 1 Pediatric Trauma Center due to bullying/child assault while at school. Concussion and Reported Assault were listed on her problem list in the Electronic Medical Record. She was discharged on the same day. The patient was identified on the ED status board by an HVIP social worker and approved to be approached by clinical staff. The social worker approached the legal guardian (mother) at the bedside. Basic demographic information was obtained. Open-ended questions were asked about the event (e.g., triggering/precipitating situation, how many classmates involved, number of violent injury incidents experienced by the victim, etc.), whether the patient felt safe at home, how the school responded, and if a police report was made. The Columbia Suicide Severity Rating Scale (CSSRS) was completed. Suicidal ideation and self-injurious behavior were denied. The HVIP was explained, as were the case management and trauma-informed services/interventions that could be provided over the course of one year. The intake packet was reviewed, which consisted of psychoeducational material on child trauma from the National Child Traumatic Stress Network (NCTSN), a Victims of Crime Application (VOCA), and consent forms. After providing consent for services, a follow-up appointment to conduct the assessment was scheduled. The patient/guardian had the choice of being seen in an outpatient facility within the hospital, their home, and/or within the community (e.g., library).

### 3.2. Assessment

The assessment was conducted over the course of 2 one-hour sessions by a Licensed Independent Social Worker (LISW) in the patient’s home. The assessment consisted of obtaining demographic information; current social, emotional, and behavioral functioning; social determinants of health screening; as well as administration of parent and child questionnaires including the ACE-Q (parent form) and the Child and Youth Resilience Measure (CYRM). Although the CSSRS was negative, suicidal ideation was assessed and a mental health status exam was completed at each point of contact.

The patient’s mother participated in the assessment. The mother reported having a high school education, being currently unemployed, having five family members in the household, and frequently moving in the past 3 years. The median household income for the patient’s zip code is 40,563 USD. No concerns for food insecurity, clothing, or shelter were reported. Social determinant screening indicated that, while the family has their basic needs consistently and adequately met, they struggle financially. An initial assessment showed no known history of child abuse or neglect, parent incarceration, or parent mental illness. A psychosocial assessment indicated that the patient had experienced both direct (i.e., verbal aggression) and indirect (i.e., group exclusion) bullying at previous schools and at her current school by the classmates who assaulted her. Per the patient’s report, she is unsure what triggered the assault. This is the first assault she has experienced. The patient believes she has been a target for bullying due to being multiracial and because of her physical appearance. Protective factors include the patient being motivated to do well in school and a supportive family network.

The social worker, caregiver, and patient collaboratively created an individual service plan (ISP) to act as a roadmap for treatment. Goals included improved academic performance and interpersonal relationships, as well as decreased psychological distress associated with the assault. Services for intervention included case management (e.g., referrals for basic needs items and services, transportation, linkage to positive activities, and developmental opportunities). An overview of Psychological First Aid (PFA) and Skills for Psychological Recovery (SPR) was provided as these modular approaches are designed to decrease distress in the immediate aftermath of a traumatic event.

### 3.3. Case Conceptualization

The patient is a 12-year-old female self-identified as multiracial living in a single family home in an urban setting. She was assaulted by multiple same-age females in the classroom. The patient was punched and kicked, had her hair pulled, and was pushed into desks. She has a significant medical history for a cranial birth defect that required operative intervention. She has no other significant medical or psychiatric history. The patient has a history of exposure to adverse childhood experiences including being bullied in multiple school settings for the past 3 years, lack of paternal involvement, financial instability, and residing in a high-crime neighborhood. Current stressors include recently moving from the suburbs to the city, birth of a sibling, and the murder of a close family member. Protective factors include a strong relationship with her mother and maternal uncle, motivation to do well academically, and a history of participation in extracurricular activities. Presenting concerns include increased emotionality, negative affect, interpersonal conflict (e.g., sibling arguing), aggressive behavior, and behavioral problems in the school setting (e.g., talking back to teachers resulting in disciplinary action). The patient reported intrusive thoughts/memories, negative mood and perception of the future, and hypervigilance associated with the current assault and because of a history of bullying.

## 4. Results

### 4.1. Pre-Treatment Data

The Adverse Childhood Experience Questionnaire (ACE-Q) [9] was completed by the guardian with the caregiver responding on behalf of the child. The ACE-Q is a 20-item questionnaire that checks for a patient’s exposure to childhood psychological, physical, and sexual abuse, as well as household dysfunction including domestic violence, and parent substance use and incarceration. This scale has been shown to have acceptable internal consistency in a variety of samples and satisfactory convergent validity with the Childhood Trauma Questionnaire [10]. An increased number of ACEs is associated with higher levels of perceived stress and mental health problems. Scores range from 0 to 20, with scores greater than 4 being considered clinically significant. The patient had a score of 8 based upon how the caregiver endorsed items.

The Child Youth Resiliency Measure (CYRM) Self-Report Form and the Person Most Knowledgeable—Child Youth Resiliency Measure (PMK-CYRM) [11] were completed by both the patient and the caregiver to measure social-ecological resilience. The overall score comprised two subscales: Personal Resilience and Caregiver Resilience. The CYRM consists of 17 questions with a three-point Likert-type response scale, with higher scores indicating greater resilience. The minimum score is 17, and the maximum score is 51. For the Personal Resilience subscale, scores range from 10 to 30, and for the Caregiver Resilience subscale, scores range from 7 to 21. This scale has been used in 14 countries and has demonstrated good reliability and validity [12,13]. Both the patient-reported overall CYRM score (Total score = 45) and the caregiver-reported overall CYRM score (Total score = 44) indicated that the child had high overall resilience. The CYRM child self-reported personal resilience score (Subscale score = 25) and parent-reported resilience score (Subscale score = 23) indicated that their perceptions of the child’s resilience were similar and serve as a protective factor for both the patient and the caregiver. The caregiver and the patient did indicate, however, that they did not feel that the child was accepted by her school. Moreover, the caregiver indicated that she did not feel that her child was treated fairly in her community.

### 4.2. Intervention

The AI team worked with research sociologists with specialization in public health and policy to develop an engagement and treatment model that provides an overview of how services are delivered to move patients from physical recovery to engagement, followed by resilience and ultimately leading to posttraumatic growth. See Figure 1.

Using the intensive case management model as an overall treatment guide, the family progressed through the treatment phases of Engagement, Assessment, Planning, Intervention, Monitoring, and Termination. In AI, social workers provide two types of services—case management and trauma-informed counseling. Participating families may receive only one service or may receive both services depending upon their needs and their choice. This family received both case management and counseling services. Patient needs are identified during the Assessment phase. Goals are created and agreed upon during the Treatment Planning phase. Needs are categorized by the following domains: Safety, Health, Basic Needs (food and clothing), Housing, Substance Use/Abuse, Education, Interpersonal Relations, Legal, Employment/Vocational, Cultural, and Other. This patient was given services in alignment with her identified needs (education and interpersonal relations). The AI social worker assisted with advocacy in her current school setting and aided in the transition to a new school. The patient was also given resources for participation in extracurricular activities to increase a sense of belongingness at her new school. Community-based programs were also identified that aligned with her interests in the medical field.

Psychological First Aid (PFA) [14] was initially provided. PFA is an empirically supported modular approach that is built on the concept of promoting human resiliency and reducing stress symptoms in response to a traumatic event. PFA was used in the Engagement phase and initially in the Resilience phase to establish safety in the school setting (advocacy; development of a school safety plan), to provide psychoeducation on trauma, Y promote self-empowerment, and to connect the patient with social supports (e.g., school-based counselor). Three PFA modules were delivered—Safety and Comfort, Connection with Social Supports, and Information on Coping. Services provided in this phase of intervention related to establishing safety in the school setting; supporting and strengthening relationships with all immediate family members, new classmates, and new peers in extracurricular activities; and introduction to basic coping skills (e.g., diaphragmatic breathing) and promotion of adaptive behavioral functioning (e.g., establishing routine). Skills for Psychological Recovery (SPR) [15] was used in the Resilience phase. SPR is an empirically supported modular secondary prevention model implemented after PFA that uses skill-building components to decrease the physical, psychological, and behavioral effects of experiencing a traumatic event. The specific SPR modules that were provided include Building Problem Solving Skills (i.e., to improve academic performance and interpersonal relationships at school), Positive Activity Engagement (i.e., to promote connectedness and to improve mood and confidence), Promoting Helpful Thinking, and Rebuilding Healthy Social connections.

To support engagement in positive activities, the social worker wrote a letter of recommendation to support the patient’s participation in a summer internship program that provides peer support for young people of color who are seeking to become physicians. To promote helpful thinking, the framework of being a “good coach, bad coach” was used to demonstrate the connection between thoughts and feelings and to identify unhelpful automatic thoughts related to interaction with peers and the perception others might have of her, and how to replace those negative thoughts with alternate helpful thoughts generated by identifying evidence for and against the maladaptive thought. The social worker also provided ongoing support to help the patient build connections at her new school. After session 6, the patient completed the Child PTSD Symptom Screen for DSM V Self Report (CPSS-V SR). The patient had a score of 20 indicating “mild” symptoms. Six additional sessions were provided, with the patient graduating from the program after a total of 12 sessions. Intervention was provided over the course of 9 months.

### 4.3. Post-Treatment Data

Post-treatment data were obtained 9 months after pre-treatment. The results on the CYRM showed that the overall child-reported score increased by one point (CYRM Overall Child Report Total Score = 46), with the overall caregiver-reported score also increasing by one point (CYRM Overall Parent Report Total Score = 45). The increase in the Child Self-Report form was due to positive endorsement of “I feel that I belong/belonged at my school”. The increase in the Caregiver Report form was due to positive endorsement of “They are treated fairly in their community” and “They feel they belong at their school”. In the subcategories of Personal Resilience, the child-reported personal resilience score increased one point (CYRM Child Self-Report Personal Resilience Score = 26) and the caregiver-reported personal resilience score increased by 2 points (Parent Report Personal Resilience Score = 25). In the subcategories of Caregiver Resilience the child self-reported caregiver resilience score remained the same (CYRM Child Self-Report Caregiver Resilience Score = 20) and the parent-reported caregiver resilience score decreased by one point (Caregiver Report Caregiver Resilience Score = 20).

The Post Traumatic Growth Inventory (PTGI) [16] was completed by both the patient and the caregiver upon treatment termination. The PTGI is used to identify clinically significant levels of posttraumatic growth in the areas of new possibilities, relating to others, personal strength, spiritual change, and appreciation of life. The PTGI has 21 questions with responses provided on a four-point Likert-type scale with scores ranging from 21 to 84, with higher scores indicating greater posttraumatic growth. The PTGI has strong internal reliability and support for content validity [17]. The overall child-reported score was 61, and the overall caregiver-reported score was 40. Interestingly, the patient reported some degree of positive gain in every category. Most significantly, the patient reported that she was “better able to understand other people’s feelings”, “able to do a better job on work at home and in school”, “had the chance to do things she wouldn’t have been able to do before”, “Learned that sometimes she needs someone to help her out”, and that she “Has new ideas about how she want things to be when she grows up”. The caregiver reported that the most significant changes were that “New opportunities are available which would not have been available otherwise” and “They have more compassion for others”. At the time of discharge, the patient experienced fewer intrusive thoughts/memories associated with bullying, a reduction in traumatic symptoms, improved mood, increased hopefulness, and engagement in positive activities (e.g., internship program; summer camp). The patient had an increase in sense of belonging at both her new school and with other children of color in her desire to be a physician. She also had a newly identified interest, learning to speak her mother’s native language of Spanish, reflecting her increased sense of cultural identity. These changes in particular represent posttraumatic growth in the areas of both new possibilities and relating to others.

## 5. Discussion and Conclusions

This case example demonstrates the importance of taking a patient-centered approach informed by a treatment conceptualization model to guide intervention. Such an approach builds on the patient’s individual goals and existing protective factors. Intensive case management, PFA, and SPR were effective with this patient in decreasing distress and promoting resilience. This study demonstrates the application of the AI model in conceptualizing the transformation from physical recovery to posttraumatic growth. Several considerations must be made. This patient had family support, which serves as a significant buffer against community violence. Although this patient had a clinically meaningful ACE score pre-treatment, she also had a high resilience score. Previous exposure to adverse childhood experiences is common in patients who receive services from HVIPs. It is important to consider how a high ACE score in combination with a low resilience score might impact program engagement, treatment, and retention.

The vast majority of studies evaluating the efficacy of HVIPs focus on an adult population. It is important to note that the research from various programs across the United States (e.g., San Francisco’s Wraparound Project; Philadelphia’s Healing Hurt People) indicate that the most commonly identified need by patients is mental health services [18]. Current studies are underway to identify the most cost-effective model for screening and risk stratifying patients who are likely in need of mental health services to address posttraumatic stress symptoms after violent injury [19]. Two randomized control trials that focused on the pediatric population showed that there was minimal effect on decreasing fighting-related injuries and behavior [20,21]. This underscores the importance of additional research needing to be conducted with larger sample sizes and within the pediatric population. Moreover, additional studies should be conducted on those who are victims of violent injury rather than on aggressor–victims. A strength of the AI model is that it can be used in response to a variety of violent injuries with the goal of posttraumatic growth regardless of the patient’s role in the conflict.

In sum, HVIPs share a common mission to mitigate the emotional effects of community violence and to reduce the risk of violent re-injury by supporting survivors of peer and community violence as they recover. Although there is variability in assessment and treatment provided by HVIPS, and no standardized approach has been proposed, this paper identifies key assessment and trauma-informed practices that can be implemented within the context of the posttraumatic growth model. Based on the experiences of this HVIP, it is recommended that social needs, previous exposure to childhood adversities, and protective factors/sources of resilience be assessed, as well as ongoing assessment of posttraumatic stress symptoms and completion of mental health status exams. Trauma-informed treatment approaches such as PFA and SPR can be helpful in decreasing a patient’s distress, in building on current strengths and supports, and in learning and applying coping skills. A quasi-experimental study using the proposed AI model should be conducted to assess its application for decreasing posttraumatic stress symptoms and building posttraumatic growth with various types of injuries and across diverse populations and ED settings. Future research should also focus on identifying the most cost-effective delivery mechanisms for both prevention and intervention that can be delivered to high-risk youth presenting to the ED.

## Figures and Tables

**Figure 1 children-10-01018-f001:**
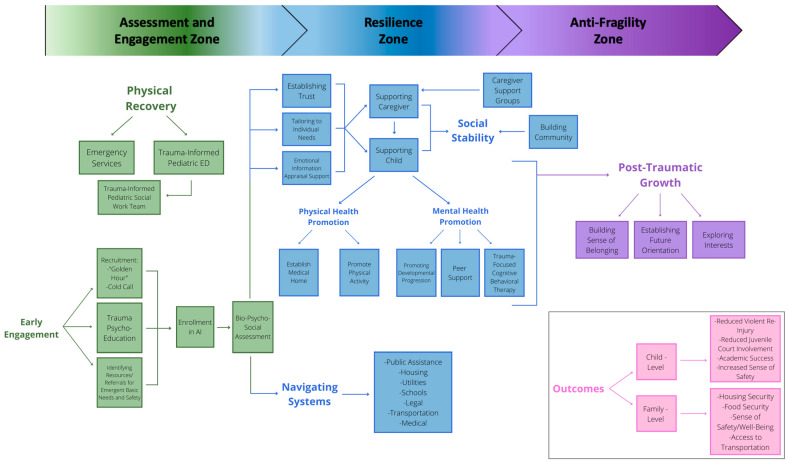
Conceptualization of the Antifragility Model (AI) within a pediatric. Emergency Department (ED).

## Data Availability

Not applicable.

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
