# Peer review of "Psychosocial Treatment of an Adolescent in a Hospital-Based Violence Intervention Program"

_children, 2023, doi:10.3390/children10061018_

Round 1
Reviewer 1 Report
1.- In the description of the case, more information should be added about the harassment suffered, whether it was a single episode or more and what the immediate triggers were.
2.- It would be convenient to give the psychometric characteristics of the evaluation instruments used. as well as its date and author
3.- An outline of the procedure is missing: Evaluation time. Duration of treatment. Reassessment time. Some of the data appears in the text, but the procedure is not formalized.
4.-It would be convenient for a greater visualization of the effects of the treatment, to add a table or figure where the scores of both the adolescent and her caregiver before and after the treatment are displayed.
Author Response
May 24,2023
Thank you for your review of “Psychosocial Treatment of an Adolescent in a Hospital-Based Violence Intervention Program”. We have incorporated your suggestions to enhance our manuscript. We appreciate the opportunity to revise the paper for further review for the Special Issue (ACEs: Prevention, Intervention, and Access to Care) in Children.
Reviewer 1
Comment: More information should be added about the harassment suffered.
Response: Additional information has been provided in the 3.2 Assessment section: “Psychosocial assessment indicated that the patient had experienced both direct (i.e. verbal aggression) and indirect bullying (i.e. group exclusion) at previous schools and at her current school by the classmates who assaulted her. Per the patient’s report, she is unsure what triggered the assault. This is the first assault she has experienced. The patient believes she has been a target for bullying due to being multiracial and because of her physical appearance.” We also indicated the duration of bullying in the 3.3 Case Conceptualization section.
Comment: It would be convenient to give the psychometric characteristics of the evaluation instruments, as well as date and author.
Response: We have included the psychometric properties of all instruments used in the evaluation. The instrument (author and date) and validation studies are cited in the References.
Comment: An outline of the procedure is missing.
Response: We added a procedure section. Additional information is provided throughout the manuscript related to the duration of the assessment, follow-up sessions, and program length.
Comment: It would be convenient for a greater visualization of the effects of the treatment to add a table of questionnaire scores.
Response: We appreciate this suggestion. However, providing the scores within the text of the manuscript allows for increased conceptualization of the scores in the context of the treatment.
Reviewer 2 Report
Dear authors, thank you for giving me the opportunity to review your manuscript: “Psychosocial Treatment of an Adolescent in a Hospital-Based Violence Intervention Program”.
I send below some comments to the manuscript.
Introduction:
- Line 22-32: The authors briefly reference interpersonal violence and its impact in a few studies. This part should be extensively developed using recent empirical studies.
- The Introduction does not present any theoretical basis on how the literature validates the treatment of adolescent victims of violence. It does not present what should be worked on, the importance of this work, etc.
- The introduction should be completely reformulated, presenting empirical evidence of what has been carried out in intervention programs with these young people.
3.2. Intervention
- This section does not provide enough information. This section should be described in more detail.
Discussion and Conclusions
- The discussion only indicates results and does not discuss them based on the literature. It is not a discussion.
Overall, the paper lacks much investment.

Author Response
May 24,2023
Thank you for your review of “Psychosocial Treatment of an Adolescent in a Hospital-Based Violence Intervention Program”. We have incorporated your suggestions to enhance our manuscript. We appreciate the opportunity to revise the paper for further review for the Special Issue (ACEs: Prevention, Intervention, and Access to Care) in Children.
Reviewer 2:
Comment: In the Introduction, the authors briefly reference interpersonal violence and its impact in a few studies. This part should be further developed.
Response: The introduction has been revised. Additional studies are referenced to support that interpersonal violence is a serious public health concern and to demonstrate limited studies evaluating outcomes for pediatric HVIPs.
Comment: The Introduction doesn’t present any theoretical basis.
Response: We elaborated on the posttraumatic growth theory in which the AI program is built. Revision of the Introduction includes the uniqueness of this case study and gaps within the literature.
Comment: The Introduction should be completely reformulated.
Response: The Introduction was revised.
Comment: Intervention – This section doesn’t provide enough information. This section should be described in more detail.
Response: A Procedure section has been added which provides additional information about the services received. Additional information has been added related to the specific interventions provided (Psychological First Aid and Skills for Psychological Recovery) as well as the patient’s response to the intervention.
Comment: The discussion indicates results and does not discuss them within the context of the literature.
Response: Reference to studies that relate to and support findings has been included in the Discussion.
Reviewer 3 Report
1.In abstract section, “ED” should be defined clearly.
2.In figure 1, AI and ED should be denoted below the figure for clarification.
3.The discussion section had no reference to discuss?
4. please describe how this manuscript referred to “CARE guidelines”.
Author Response
May 24,2023
Thank you for your review of “Psychosocial Treatment of an Adolescent in a Hospital-Based Violence Intervention Program”. We have incorporated your suggestions to enhance our manuscript. We appreciate the opportunity to revise the paper for further review for the Special Issue (ACEs: Prevention, Intervention, and Access to Care) in Children.
Reviewer 3:
Comment: In the abstract, ED should be clearly defined.
Response: This has been included.
Comment: In figure 1, AI and ED should be clarified.
Response: This has been done.
Comment: The discussion section had no references to discuss.
References have been added.
Comment: Please describe how this manuscript referred to CARE guidelines.
Response: We have detailed below how the manuscript follows CARE guidelines, as well as included a statement within the manuscript.
The authors referred to the CARE guidelines as indicated below.
Abstract: Additional information has been provided on the uniqueness of the case.
Introduction: Added/emphasized why this case is unique and what it adds to the literature (e.g. “This paper fills an important gap in the literature by providing a conceptualization model for pediatric HVIP intervention and by demonstrating how empirically-supported practices can be delivered within the context of the model. This case study is intended to serve as a guide for HVIPs by outlining the services provided and subsequent treatment trajectory of an adolescent receiving care from a HVIP operating out of a Level 1 Pediatric Trauma Center in the Midwest”).
Patient Information: Included within the manuscript are specific de-identified data, primary patient concerns, and relevant medical and psychosocial history. The patient had not received previous behavioral health intervention.
Clinical Findings/Timeline/Diagnostic Assessment: Included under Case Presentation and Results
Therapeutic Intervention: This is identified within the manuscript (Psychological First Aid and Skills for Psychological Recovery)
Outcomes: Included within the manuscript.
Discussion: Added strengths and limitations of the case, relevant literature, and primary takeaway.
Round 2
Reviewer 2 Report
The authors have invested in the article, fixing the issues presented in the first version.
Reviewer 3 Report
OK. The authors have finished the related suggestions.